# Research Progress on the Therapeutic Effect of Polysaccharides on Non-Alcoholic Fatty Liver Disease through the Regulation of the Gut–Liver Axis

**DOI:** 10.3390/ijms231911710

**Published:** 2022-10-03

**Authors:** Xiang Chen, Menghan Liu, Jun Tang, Ning Wang, Yibin Feng, Haotian Ma

**Affiliations:** 1Key Laboratory of Combinatorial Biosynthesis and Drug Discovery, Ministry of Education, School of Pharmaceutical Sciences, Wuhan University, Wuhan 430071, China; 2School of Chinese Medicine, The University of Hong Kong, 10 Sassoon Road, Pokfulam, Hong Kong 999077, China

**Keywords:** polysaccharide, gut–liver axis, NAFLD, gut microbiota, intestinal barrier

## Abstract

Non-alcoholic fatty liver disease (NAFLD) is the most common chronic liver disease affecting global public health at present, which can induce cirrhosis and liver cancer in serious cases. However, NAFLD is a multifactorial disease, and there is still a lack of research on its mechanism and therapeutic strategy. With the development of the gut–liver axis theory, the association between the gut–liver axis and the pathogenesis of NAFLD has been gradually disclosed. Polysaccharides, as a kind of natural product, have the advantages of low toxicity, multi-target and multi-pathway action. It has been reported that polysaccharides can affect the gut–liver axis at multiple interrelated levels, such as maintaining the ecological balance of gut microbiota (GM), regulating the metabolites of GM and improving the intestinal barrier function, which thereby plays a protective role in NAFLD. These studies have great scientific significance in understanding NAFLD based on the gut–liver axis and developing safe and effective medical treatments. Herein, we reviewed the recent progress of polysaccharides in improving nonalcoholic fatty liver disease (NAFLD) through the gut–liver axis.

## 1. Introduction

Non-alcoholic fatty liver disease, which mainly refers to a clinicopathological syndrome characterized by lipid accumulation and inflammation, is caused by multiple factors (insulin resistance, genetic predisposition, improper diet, obesity), in addition to alcohol and other certain liver damage factors (physical, viral, chemical liver damage). It includes non-alcoholic fatty liver (NAFL), non-alcoholic steatohepatitis (NASH), related liver cirrhosis and hepatocellular carcinoma [1]. Over the past 20 years, urbanization and rising living standards in many Asian countries have led to sedentary lifestyles and over-nutrition, which have contributed to obesity and NAFLD epidemics [2]. As of 2018, the global prevalence of NAFLD was reported to be approximately 24% [3], while the prevalence of NAFLD in China reached 29.2% [4]. Despite the high morbidity, there are still many deficiencies in our understanding of the pathogenesis, prevention, diagnosis, and treatment of NAFLD.

In recent years, more and more researchers have focused on the study of the gut–liver axis and found that the intestinal mucosal barrier, gut microbiota (GM) and its metabolites are closely related to the pathogenesis of NAFLD [5]. The gut–liver axis refers to the bidirectional relationship between the gut and its microbiota and the liver, which is a complex of signals caused and influenced by multiple factors including genetics, diet, and the environment [6]. Exploring the pathogenesis of NAFLD based on the gut–liver axis and developing safe and effective therapeutic drugs has become important directions of chronic liver disease research in recent years [7].

Polysaccharides are biological macromolecules composed of 10 or more monosac-charides linked by glycosidic bonds. Structurally, glycosidic bonds can link different types of monosaccharide molecules in multiple ways and at multiple sites, and finally form polysaccharides with complex and diverse main and branched structures [8]. In addition, the spatial configuration of sugar chains is not unique and universal. Due to the characteristics of its composition, it can form spherical, rod-shaped, helical chains, flexible chains and other configurations [9]. In terms of physiological activity, polysaccharides with rich structural information participate in a wide range of physiological metabolic activities, providing researchers with different perspectives. A variety of dietary polysaccharides (e.g., plant polysaccharides, animal polysaccharides, fungal polysaccharides) in the human diet cannot be directly absorbed by the intestinal tract, and can only be utilized after being metabolized by a variety of GM enzymes [10,11]. The polysaccharides discussed here are different from ordinary starch or can perhaps be called non-starch polysaccharides to some extent, which can play an important role in human health as prebiotics [11]. Polysaccharides and their metabolites can produce various therapeutic effects on the body through the regulation of GM. For example, polysaccharides can play a protective role in the intestine and liver, by alleviating intestinal barrier damage [12,13], regulating lipid metabolism [14,15], improving oxidative stress [16], reducing insulin resistance [17], and inhibiting pro-inflammatory response [18]. These studies revealed the multifaceted interaction between polysaccharides and GM and suggested that polysaccharides may interfere with the occurrence and development of NAFLD by regulating the gut–liver axis. Accordingly, this paper summarized the NAFLD-inducing mechanism targeting the gut–liver axis, further reviewed the research progress of polysaccharides in improving NAFLD through the gut–liver axis, and discussed its underlying mechanism as well as the prospects for the development of new therapeutic drugs.

## 2. The Pathogenesis of NAFLD Based on the Gut–Liver Axis

The gut–liver axis, consisting of the gut barrier, GM, and liver, is named after the close anatomical and biological function between the gut and liver. From an anatomical point of view, both the intestinal and liver embryos originate from the foregut. They also build a bidirectional interaction through the portal vein and the enterohepatic circulation of bile, where the portal vein can transport intestinal-derived products to the liver, and then the liver secretes bile and antibodies into the gut. In terms of biological functions, the gut and liver together constitute a defense barrier for the body to contact exogenous substances [19]. Therefore, the body needs to maintain the homeostasis of the gut–liver axis.

The pathogenesis of NAFLD is complex, and previous studies have focused on de novo lipid synthesis in the liver and oxidative stress responses [20,21]. With the proposal of the gut–liver axis raised, the rise of GM research, and the in-depth study of human pathology, researchers began to systematically link signaling pathways associated with the gut–liver axis to the mechanisms of liver injury in NAFLD. A theory about multiple hits leading to NAFLD involving gut–liver axis disorder had been widely noticed [21]. Studies have shown that gut–liver axis dysfunction caused by dysbiosis in the gut–liver axis and intestinal barrier damage were important factors leading to the development of NAFLD [22]. If the intestinal flora were imbalanced in the body, the intestinal epithelial barrier function would be damaged, while bacteria and GM metabolites would be transferred to the liver and trigger corresponding inflammatory immune responses. These undesirable changes further promoted the occurrence and development of NAFLD. In addition, GM metabolites, in which LPS, short-chain fatty acids (SCFAs), bile acids (BAs), and choline also directly or indirectly affected the phenotype of liver tissue, might be involved in various pathological stages such as NAFL, NASH, and liver cirrhosis [23,24]. These findings undoubtedly provided new ideas for the clinical prevention and treatment of NAFLD.

### 2.1. Intestinal Dysbiosis and NAFLD

GM is a diverse ecosystem composed of bacteria, protozoa, archaea, fungi and viruses, which also coexist with the human body in a specific symbiotic relationship [7]. The flora colonizes the human gut with diverse functions, large numbers and complex genomes, most of which cannot be artificially cultured at present. More than 90% of the bacterial components in GM are mainly composed of four phyla: Firmicutes, Bacteroidetes, Actinobacteria, and Proteobacteria [25,26]. However, due to environmental or genetic factors, the types and proportions of GM also vary greatly among individuals [27].

GM is a part of the human metabolism and immune system, which is essential for human health. Under normal circumstances, huge amounts of microbiota coexist in a certain proportion, maintain the intestinal microecological homeostasis, and participate in the process of food digestion, bile acid metabolism, and intestinal mucosal immunity [7]. Studies have shown that changes in the composition and proportion of some microbiota in patients with NAFLD may lead to the destruction of the microbial barrier formed by GM, and the dysbiosis of intestinal flora, which ultimately promoted the process of NAFLD [28,29].

A variety of specific metabolites may be involved in the interaction of GM with the liver through the gut–liver axis. Among them, LPS, secondary bile acids, SCFAs, and choline metabolites, such as trimethylamine (TMA), and trimethylamine N-oxide (TMAO), were found to be associated with NAFLD [30]. Dysbiosis of the microbiota may alter the balance and function of metabolites, which in turn affects the occurrence and development of NAFLD (Figure 1).

Under normal circumstances, the intestinal flora may conduct material exchange (e.g., LPS, SCFAs, and BAs) and signal transduction (e.g., FXR and TGR) with the liver via the intestinal barrier. When the homeostasis of the intestinal flora was disrupted, the composition and proportion of the flora may be changed, and accordingly, the metabolites of the flora may be affected. These events may eventually lead to liver inflammation, insulin resistance, lipid metabolism and glucose metabolism disorders, and further the occurrence and development of NAFLD.

(I) The concentration of LPS may increase owing to the increased producing bacteria. LPS stimulation may damage the local mucosa of the intestinal tract, which may cause LPS to pass through the compromised intestinal barrier and enter the liver through the portal vein and finally induce an inflammatory cascade in both the intestine and liver.

(II) SCFAs may participate in the regulation of lipid and energy metabolism, insulin sensitivity, and oxidative stress. When the total amount of SCFAs decreased, their positive regulatory function in the liver would be attenuated.

(III) Trimethylamine (TMA), a metabolite formed from choline metabolism by gut flora may be transformed to trimethylamine oxide (TMAO) in the liver, which may increase the level of TMAO, reduce the utilization of choline in the liver, and lead to dysregulation of hepatic lipid metabolism and glucose metabolism.

(IV) During the bile acid cycle, the bile acids converted from cholesterol in the liver may be metabolized into secondary bile acids by the intestinal flora. When bile acid homeostasis was disrupted, in that the secondary bile acids are reduced, the corresponding FXR and TGR5 signaling in the liver may be weakened, which may in turn adversely affect hepatic lipid and glucose metabolism.

LPS is a complex amphiphilic molecule derived from the cellular wall of Gram-negative bacteria, consisting of a hydrophobic part (lipid A) and a hydrophilic part (carbohydrate core and polysaccharide *O*-antigen) [31]. LPS was currently considered to be one of the major factors contributing to the occurrence and progression of NAFLD [32]. LPS may form complexes with its associated LPS-binding protein (LBP), which may be transported to membrane-bound or soluble cluster of differentiation 14 (CD14) to specifically bind to Toll-like receptors (e.g., TLR4), and subsequently induce TLR4 activation. This would in turn activate pro-inflammatory and pro-fibrotic pathways (e.g., MAPK, and NF-κB signaling pathways), then increase the expression of pro-inflammatory factors (e.g., TNF-α, IL-1, IL-6 and IL-8), and finally produce a pathological effect on the liver [33].

Bile acids (BAs) are the main components of bile. Cholesterol could be metabolized to primary bile acids in hepatocytes by cholesterol 7α-hydroxylase (CYP7A1) [34]. The BAs in the liver may enter the small intestine through the duodenum and act synergistically with other bile components, in which 95% of the BAs may be reabsorbed in the terminal ileum and reach the liver, whereas 5% are metabolized by GM to form secondary BAs, such as deoxycholic acid (DCA) and lithocholic acid (LCA) [35,36,37,38]. Studies showed that GM may play an important role in the negative feedback regulation of bile acid metabolism and its synthesis [35]. Liver injury may cause dysbiosis of the microbiota, which may disrupt bile acid synthesis, uptake, and the ratio between primary and secondary BAs. Severe NAFLD may develop into cirrhosis, which may reduce the conversion of primary BAs to secondary BAs [39,40]. Bile salt hydrolase (BSH) of *Lactobacilli*, *Bifidobacteria*, *Clostridium* and *Bacteroides* in the gut could deconjugate BAs, which may not only increase the absorption of bile acids in the gut but also enhance toxicity tolerance to bile acids and their derivatives [41,42]. In addition, GM could carry out oxidation/isomerization, 7α hydroxylation and esterification of BAs to affect the intestinal absorption and toxicity of BAs [41]. Conversely, BAs also have a greater impact on the microbiota structure. Studies showed that BAs could destroy the GM cell membrane and then cause damage to its internal structure, which may inhibit the overgrowth of bacteria [43]. Since BAs are the final products of cholesterol metabolism in the body and also an essential substance for lipid absorption, it is very important to maintain the homeostasis of BAs for the regulation of lipid metabolism and prevention of bile acid toxicity and gastrointestinal inflammation [44,45,46]. In this process, a variety of receptors, such as G protein-coupled membrane receptor 5 (TGR5, or G protein bile acid-coupled receptor 1 Gpbar1) and nuclear farnesol X receptor (FXR), may have interactions with cytokines (e.g., fibroblast growth factor 19, FGF19), participate in host bile acid metabolism, and affect glucose and lipid metabolism to complete the bile acid cycle [35]. In this cycle, FXR and TGR5 are the two main receptors that regulate the metabolism of BAs. FXR is a transcription factor that binds to the promoter region and initiates the expression of various target genes, the signaling of which may cause negative feedback, modulating the synthesis of BAs and their ability to shape the intestinal flora. FXR could also directly participate in inhibiting the overgrowth of intestinal microorganisms, thereby alleviating intestinal barrier dysfunction and improving intestinal dysbiosis [47]. Studies showed that activation of FXR distributed in liver tissue could increase insulin sensitivity, and reduce both obesity and the release of inflammatory factors, showing an improvement in NAFLD [48]. TGR5, a BA-sensitive G protein-coupled receptor, is widely expressed in vivo. TGR5 knockout mice were found to be more susceptible to LCA-induced liver injury, as evidenced by increased serum AST levels, biliary infarction, and elevated portal perfusion pressure [49]. It was also found that TGR5 was a negative regulator of NF-κB-mediated inflammation and could maintain glucose homeostasis in obese mice, thereby reducing obesity and hepatic steatosis [50,51]. In addition, FXR also plays an important regulatory role in the BAs’ enterohepatic circulation and their transportation from the liver to the intestine. In hepatocytes, FXR induces the expression of transporters such as bile salt export pumps (BSEP or ABCB11) to discharge the BAs into bile; on the sinusoidal membrane, BAs could inhibit sodium/taurocholic acid co-transporting polypeptides (NTCPs) and organic anion-transporting peptides (OATPs). Therefore, FXR may act as a bile acid sensor to maintain lower intrahepatic bile acid concentrations and prevent cholestatic liver injury [44]. It can be seen that the signaling changes of FXR, TGR5 and BAs transporters involved in bile acid metabolism are closely related to the occurrence and development of NAFLD. It is suggested that the prevention and treatment of NAFLD could be achieved by regulating bile acid metabolism and its related signaling pathways.

SCFAs, another kind of important metabolite of GM, are mainly produced in the cecum and proximal colon [52]. SCFAs is a general term for organic fatty acids with less than six carbon atoms [10], including formate, acetate, propionate, butyrate, valerate and caproate. In adult colon and feces, the three highest SCFAs are acetate, propionate, and butyrate, accounting for 95% of the total SCFAs [53]. SCFAs may not only provide energy for the intestinal epithelium, but also have a variety of biological activities, such as regulating oxidative stress, immunity, lipid and glucose metabolism, and maintaining the homeostasis of intestinal flora. This evidence indicated that SCFAs may play a positive regulatory role in the process of NAFLD [54]. The type and concentration of SCFAs in the intestinal and portal circulation may also be closely related to the progression of NAFLD [55]. One of the studies showed that three SCFAs differed significantly in their potential impact on host physiology, in which butyrate was the main energy source for colon cells, propionate contributed to hepatic gluconeogenesis, and acetic acid was an important substrate for cholesterol and fatty acid synthesis. Another research study found that dietary supplements supplemented with butyrate were effective in preventing high-fat diet (HFD)-induced obesity, insulin resistance, and hepatic steatosis, and slowing the progression of NAFLD disease [56]. Sonia et al. [57] explored the clinical characteristics of children with NAFLD through multi-factor experiments, the results of which showed that the content of formic acid, acetic acid, and valeric acid in the feces of NAFLD children was lower than that of healthy people.

Under normal circumstances, the liver uses choline to produce phosphatidylcholine (lecithin). When the microbial flora proliferates abnormally, the choline in the diet may be consumed excessively, which may reduce the effective utilization of choline by the liver and further induce dyslipidemia, eventually NAFLD. On the other hand, GM may metabolize choline into toxic TMA. Then, TMA may be oxidized by hepatic flavin-containing monooxygenase 3 (FMO3) to TMAO after being transported to the liver [58]. Once the level of TMAO is too high, it may cause the blockage of FXR signaling and the abnormal metabolism of BAs, thereby aggravating hepatic steatosis, leading to NAFLD and even NASH [59,60]. Figure 1 shows the pathogenesis of NAFLD mediated by intestinal dysbiosis.

### 2.2. Gut Barrier Permeability and NAFLD

The intestinal mucosa is a semipermeable membrane barrier with complex functions, which may serve as a line of defense for gut–liver interactions. On the one hand, intestinal mucosa allows nutrients to enter the circulatory system and reach the liver; on the other hand, it can separate the substances in the intestinal lumen and limit the transmission of intestinal microorganisms and endotoxins to the liver. Therefore, intestinal mucosa plays an important role in the homeostasis of the gut–liver axis. Broadly, the intestinal barrier is divided into the biological barrier, mechanical barrier, chemical barrier and immune barrier. Among them, the biological barrier is composed of normal GM, while the chemical barrier is composed of gastric juice, bile, mucus, mucin, mucopolysaccharide, various digestive enzymes, lysozyme, and other chemicals secreted by the gastrointestinal tract [61]. The intestinal mucosal mechanical barrier (physical barrier) and immune barrier are crucial for the maintenance and alteration of intestinal permeability. The change in intestinal permeability also greatly affects the occurrence and development of NAFLD [33].

The intestinal mucosal mechanical barrier is mainly composed of mucosal epithelial cells and their intercellular junction complexes, the latter mainly including tight junctions (TJs) and subsequent adhesive junctions. The proteins that constitute TJs include occludin and claudins, which maintain the integrity of the mechanical barrier [62]. TJs are the main intercellular junctions regulating intestinal permeability while claudins are the major skeletal proteins that make up TJs, in which the regulation of claudin-4 is essential for tight junction function. The abnormal expression of TJ proteins may cause damage to the intestinal mucosal mechanical barrier function, which may increase intestinal permeability, and promote bacterial translocation and LPS entry into the blood in NAFLD [63]. When the gut microbiome is out of balance, the LPS-producing bacteria in the intestinal lumen may increase and disrupt the intestinal mechanical barrier, inducing an increase in intestinal epithelial cell gaps and damage to the tight junctions [64]. Meanwhile, LPS may pass through the mucous and epithelial cell layers, and then enter the liver through the portal vein, which may further induce liver inflammation and promote the occurrence of NAFLD [33].

In terms of the immune barrier, lymphoid tissue and some immune cells (e.g., T cells, B cells) at the lamina propria of the intestinal mucosa together with the immune molecules (e.g., secretory immunoglobulin A, sIgA) secreted by plasma cells may act as a line of defense to enhance intestinal barrier function. This defense line may exert immune defense against invading antigens (e.g., harmful bacteria and their metabolites) that cross the intestinal mechanical barrier [65]. Studies have shown that CD4^+^ T lymphocytes in mesenteric lymph nodes (MLN) play an important role in resisting gut bacteria. Their different cell subtypes serve different functions, and the changes in the proportion of cells of their subtypes are related to NAFLD. For example, T helper cells (Th1 and Th17) derived from naive T cells are involved in the pro-inflammatory response, whereas Th2 and Treg cells have an antagonistic function [66,67]. Su et al. [67] explored the role of CD4^+^ T lymphocytes in MLN in the progression of NAFLD. They found that Th1/Th2 and Th17/Treg ratios were increased in the intestinal MLN of NAFLD mice. These results suggested the imbalance between pro-inflammatory and anti-inflammatory responses causes intestinal inflammation during the pathogenesis of NAFLD, which may further aggravate intestinal barrier damage and promote the disease process. In addition to changes in CD4^+^ T lymphocytes and their subtypes in MLN, changes in sIgA produced by B cell differentiation also affected the occurrence and development of NAFLD. IgA is the major immunoglobulin subtype in the gut which may help maintain the stability of the gut homeostasis. IgA secreted into the intestinal lumen (or sIgA) has multiple functions, including controlling the composition of the microbiota and protecting the intestinal epithelium from pathogenic microbiota [68]. Yang et al. [69] found that intestinal sIgA production was reduced in NAFLD/NASH rats, suggesting that IgA had a potential function in liver disease. There were other reports that sIgA helped to protect beneficial members of GM and had a special coating ability for Gram-negative bacteria to prevent them from adhering to the intestinal wall, thereby exerting a protective function on the intestinal barrier [70,71]. Conversely, when the intestinal mucosa is damaged, the number of sIgA plasma cells decreases, which may aggravate intestinal bacterial translocation [72]. These pieces of evidence suggest that IgA is related to the severity of NAFLD. Figure 2 shows the pathogenesis of nonalcoholic fatty liver disease induced by altered intestinal barrier permeability.

In the gut–liver axis, the gut barrier generally maintains structural integrity and permeability. Under the action of multiple factors, such as the abnormal expression of tight junction proteins, diet, drugs and physiological factors (e.g., age and stress), and abnormal changes in the composition of intestinal flora, the intestinal mucosal barrier function is impaired, and intestinal permeability occurs. The change promotes the translocation of the flora and the entry of LPS into the blood, induces inflammation, and then NAFLD.

(I) Some immune cells in the lamina propria of the intestinal mucosa (e.g., T cells and B cells in the figure) and their secreted immune molecules (e.g., IgA in the figure) can serve as a line of defense to enhance the function of the intestinal barrier. When the immune system is unbalanced, the IgA secreted by plasma cells produced by B cell differentiation decreases, which will aggravate the translocation of intestinal bacteria and the entry of LPS into the blood.

(II) Helper T cells (Th1 and Th17) derived from initial T cell differentiation participate in the pro-inflammatory response, which in turn accelerates intestinal barrier damage and causes bacterial antigens to enter the liver through the intestinal barrier, eventually promoting the occurrence and development of NAFLD.

## 3. Ameliorative Effects of Polysaccharides on NAFLD via the Gut–Liver Axis

### 3.1. Ameliorative Effects of Polysaccharides on NAFLD

In recent years, polysaccharides have attracted wide-spread attention for their physiological properties, such as anti-virus [73], anti-tumor [74], anti-inflammatory [75], anti-diabetic [76], anti-liver injury [77,78,79,80,81], reducing obesity [82], and immune regulation [83]. Polysaccharides have become one of the current research hotspots owing to the prevention and treatment of liver damage in various ways (e.g., anti-inflammatory, anti-apoptotic, anti-oxidative stress) [84,85,86,87]. 

Studies have also shown that polysaccharides play a significant role in ameliorating NAFLD, a multi-factor-induced liver injury disease recently. For example, *Gastrodia elata* Blume polysaccharides could exert a protective effect on a high-fat diet (HFD)-induced NAFLD by improving liver function in mice, regulating lipid metabolism, alleviating hepatic oxidative stress injury, inhibiting inflammation, and inhibiting hepatocyte apoptosis [88]. Chicory polysaccharides could significantly alleviate high-fat diet-induced NAFLD in rats by activating AMPK, showing a strong lipid-lowering effect [89]. Lentinan can improve NAFLD by activating the peroxisome proliferator-activated receptor alpha (PPAR-α) pathway to improve hepatic steatosis and reduce oxidative stress and apoptosis [90]. It has also been reported that the hepatoprotective effect of *Lycium barbarum* polysaccharides on NAFLD is mainly reflected in the promotion of mitochondrial biogenesis, the regulation of energy balance, a reduction in intracellular lipid accumulation, and a reduction in oxidative damage in NAFLD cell models [91]. These studies provided the basis for further exploration of active polysaccharides with the effect of preventing and ameliorating NAFLD.

### 3.2. The Mechanism of Action of Polysaccharides in the Prevention and Treatment of NAFLD through the Gut–Liver Axis

Based on the above understanding of the induction mechanism of NAFLD, it is possible to further explore the active molecules that prevent and treat NAFLD through the gut–liver axis. Due to the potential role of polysaccharides in NAFLD, the effect and mechanism of polysaccharides on improving NAFLD via the gut–liver axis have been investigated in recent years based on NAFLD disease models and changes in relevant biochemical indicators and histology, as well as in the species and composition of intestinal flora. The relevant results are shown in Table 1.

#### 3.2.1. Maintaining the Ecological Balance of GM

As the two largest phyla of GM, Firmicutes and Bacteroidetes ratio (Firmicutes/Bacteroidetes Ratio, F/B) is one of the focuses of many pieces of research. Due to the complexity of the gut microbiome, the use of F/B as a biomarker for gut dysbiosis remains controversial [113]. Analysis of the F/B ratio, however, could provide important insights into the prevention and treatment of obesity-related diseases such as NAFLD [102,103,104]. Goto et al. [102] studied the improvement effect of a sulfated polysaccharide Sacran on NASH model rats and investigated the effect of the polysaccharide on the distribution of intestinal flora. The results showed that compared with the model group without Sacran, the diversity and abundance of the intestinal flora of the rats in the administration group did not change significantly, but the relative abundances of Firmicutes and Bacteroidetes were found to decrease at the phylum level, suggesting that Sacran could alleviate NASH by changing the phylum composition. Hong et al. [103] investigated the effect and mechanism of *Astragalus* polysaccharides (APS) on HFD-induced NAFLD mice by using targeted metabolomics and metagenomics. It was found that compared with the high-fat diet model group, the abundance and uniformity of intestinal bacteria in the APS group were significantly increased. Furthermore, in the APS group, the abundance of Firmicutes decreased while that of Bacteroides increased, in that the F/B ratio decreased. These results indicated that the NAFLD hepatic steatosis may be improved by APS in a microbiota-dependent manner. Similarly, Chen et al. [104] used a high-sugar and HFD-induced NAFLD mouse model to investigate the effect and mechanism of *Sarcodon aspratus* polysaccharide (SATP) in improving NAFLD based on the micro-ecological balance of gut microbiota. The results showed that the diversity of intestinal flora in NAFLD mice was reduced, and the ratio of F/B was significantly higher than that of normal mice. GM diversity increased and the F/B ratio decreased after SATP administration. Additionally, the abundance of beneficial bacteria including Lactobacillus (Firmicutes), Bacteroides and Akkermansia (Verrucomicrobia) increased significantly, indicating that the GM disorder was improved. This study also indicated that the gut microbiome is a complex formed by the interaction of various bacterial groups, not just dominated by F/B.

Therefore, GM and its gut microbiome are key components of the gut–liver axis and are closely related to the host glucose and lipid metabolism, immunity, oxidative stress and other processes [114]. Accordingly, Polysaccharides could affect GM composition and regulate intestinal microecology in many ways. The former includes the F/B ratio, and the colonization ability of beneficial bacteria (e.g., Lactobacillus, Bifidobacterium, Gram-positive cocci and other genera) and pathogenic bacteria (the gut bacteria that can cause a variety of diseases if they are out of control); the latter includes the overall community structure such as the diversity and richness of the flora.

#### 3.2.2. Regulating the Metabolites of GM

##### LPS

As previously mentioned, increased intestinal permeability resulting from impaired intestinal mechanical and immune barriers is one of the important mechanisms promoting the progression of NAFLD. LPS produced by intestinal flora metabolism might activate intestinal mucosal adenylate cyclase, affect mitochondria and lysosomes in epithelial cells, and then cause damage to the top of intestinal villi and epithelial cells [48]; on the other hand, LPS could pass through the damaged intestinal wall barrier, enter the blood through the portal vein and reach the liver, activate the TLR4 pathway, and subsequently induce an inflammatory cascade [115]. Polysaccharides could ameliorate NAFLD by reducing LPS levels or protecting the intestinal barrier, thereby inhibiting TLR4-related pathways and modulating inflammation. Wu et al. [96] studied the improving effect and mechanism of a new “α-D-glucan” (MP-A) mussel polysaccharide on NAFLD. The results showed that MP-A had a good regulatory effect on GM and its metabolites in NAFLD model rats, not only ameliorating the imbalance of intestinal flora in rats but also inhibiting the expression of TLR4 and NF-κB in hepatocytes and the expression of downstream inflammatory factors. These effects improved systemic inflammation in the ileum and liver, reduced the damage and penetration of LPS to the intestinal mucosa, and alleviated the NAFLD process under the co-regulation of other pathways (e.g., the SCFA pathway). Ye et al. [116] studied the ameliorating effect of *Poria cocos* polysaccharide (PCP) on NAFLD and explored its underlying mechanism. The results showed that PCP could effectively improve the liver injury of NAFLD. In comparison with the model group, the expressions of tight junction proteins (ZO-1 and occludin) were significantly up-regulated, while the expression of the intestinal vascular barrier (GVB) damage marker protein PV1 was significantly decreased, suggesting that PCP could protect the integrity of GVB and reduce the transfer of LPS to the liver. A further study on the mechanism found that PCP could inhibit the pyroptosis of small intestinal macrophages by regulating PARP-1 to protect the integrity of the intestinal barrier under a high-fat diet, and thus play a role in improving NAFLD.

##### SCFAs

Human small intestinal glucoamylase is suitable for digesting and degrading food-derived starch, lactose and sucrose, while the remaining complex polysaccharides (e.g., non-starch polysaccharides, resistant starch and dietary fiber) that are not digested are mainly decomposed by GM fermentation, and are the main carbon source of GM [11,117]. When polysaccharides reach the colon, GM will degrade polysaccharides into monosaccharides or oligosaccharides through different degradation systems and transport systems, and then transport these monosaccharides or oligosaccharides into the cell for further degradation and fermentation, and finally form SCFAs with 1-6 carbon atoms [10,118]. SCFAs mainly exist in the form of acetate, propionate and butyrate in the human body, which are of great benefit to the health of the host [119]. Studies have shown that SCFAs had a very positive effect on energy metabolism in mammals, and SCFAs can be used as ‘metabolic fuel’ together with glucose [120]. It has been reported that 70% of the energy required by intestinal epithelial cells (IEC), which constitute the intestinal mechanical barrier, was provided by butyric acid, one of the SCFAs, and this butyric acid or salts were mainly produced by commensal bacteria of the Firmicutes genus Clostridium (e.g., Ruminococcus and Faecalibacterium) [119]. Butyrate could be utilized by intestinal epithelial cells and absorbed by monocarboxylate transporter (MCT1) and sodium-coupled monocarboxylate transporter (SMCT1) to promote cellular metabolism. A small amount of butyrate with other SCFAs pass through the intestinal epithelium into the portal vein and enter the circulation, where they are utilized by various cells [121,122]. On the other hand, SCFAs produced by GM metabolism also played a positive regulatory role in NAFLD [122,123]. Sun et al. [105] found that an insoluble polysaccharide WIP in the sclerotia of Poria cocos could significantly increase the content of butyric acid in the feces of NAFLD mice, and butyrate could stimulate the release of mucin and increase the ileal mucosal integrity protein (Muc-5) and the transcriptional expression of tight junction proteins (ZO-1 and occludin) and maintain the integrity of the intestinal barrier, thereby ensuring normal intestinal barrier function and preventing harmful bacteria from entering the liver through the intestinal barrier to cause NAFLD. However, the production of these ‘fatty acids’ requires appropriate fermentation processes and corresponding substrates or prebiotics (e.g., dietary fiber and non-starch polysaccharides) [117,124]. Moreover, there are not many types of enzymes with which the human body can digest and break down these polysaccharides. At present, there are only 17 known enzymes in the human genome that can encode polysaccharides in food, such as Alpha-amylase, Pancreatic alpha-amylase, and Lactase-phlorizin hydrolase [125,126]. After entering the gut through dietary intake, polysaccharides are degraded by carbohydrate-active enzymes (CAZymes), which are abundant in GM, and further form SCFAs [125]. Thus, the production of SCFAs depends not only on the composition and abundance of the relevant flora in the GM but also on whether the provided food contains an appropriate ‘fermentation substrate’. Studies have shown that polysaccharides could not only increase the colonization ability of SCFAs-producing bacteria (Lactobacillus, Lactococcus, Allobaculum, Butyricimonas) by regulating the GM microecology [97], but also may be metabolized as GM substrates to form SCFAs in the cecum and colon of mice [95]. Takayama et al. [107] reported that a partially hydrolyzed guar gum (galactomannan, PHGG) ameliorated NAFLD induced by increased intestinal permeability combined with atherogenic administration, while PHGG significantly increased the abundance of butyric acid-producing bacteria in the gut (e.g., *Clostridium* subcluster XIVa), suggesting that the polysaccharide mediate the production of SCFAs, play a role in protecting the intestinal barrier function, and ultimately achieve the purpose of alleviating NAFLD. From the perspective of source, Hu et al. [127] used psyllium polysaccharide as material to study the relationship between the consumption of different types of monosaccharides in the polysaccharide and the formation of main SCFAs, and the results showed that acetate and butyrate were mainly derived from glucuronic acid and xylose by GM fermentation, while propionic acid was derived from arabinose and xylose, suggesting that the monosaccharide composition of the polysaccharide may determine the type of SCFAs produced in the gut.

SCFAs are not only an essential energy source for the body but also function as signaling molecules. In recent years, the role of SCFAs in lipid metabolism, inflammation, and oxidative stress has attracted the attention of many scholars. Yao et al. [128] found that SCFAs could bind to endogenous SCFAs receptors (e.g., G protein-coupled receptor 43, GPR43) in the liver, and had extensive effects on the body’s physiological functions through various signaling pathways. Polysaccharides can alleviate NAFLD through SCFAs, and this is mainly reflected in the regulation of hepatic lipid metabolism [96,97,129]. Besten et al. [129] found that dietary supplementation with SCFAs could selectively inhibit the expression and activity of PPARγ and activate the UCP2-AMPK-ACC pathway, in which the phosphorylation of AMPK can inactivate the ACC enzyme, resulting in reduced malonyl-CoA production and the activity of CPT-1 (Carnitine palmitoyltransferase I) increases, which in turn promotes the β-oxidation of fatty acids, reduces lipid accumulation in the liver, and finally prevents and reverses the metabolic abnormalities induced by a high-fat diet. Wu et al. [96] also detected the concentration of SCFAs in the cecal contents by gas chromatography and analyzed the transcription and protein levels of PPARγ and SREBP-1c in the liver, respectively. The results indicated that MP-A enhanced the production of SCFAs in the gut while inhibiting the expression and activity of PPARγ and SREBP-1c, thereby counteracting HFD-induced NAFLD.

##### Bile acids (Primary and Secondary BAs)

Cholesterol forms BAs through the action of enzymes such as cholesterol synthesis rate-limiting enzyme CYP7A1 in the liver, and this amphiphilic molecule is released in the biliary tract after binding to glycine or taurine. In the intestine, BAs are involved in the digestion of emulsified fat, cholesterol, and fat-soluble vitamins, and promote the absorption of meals in the small intestine. Although most BAs are reabsorbed back into the liver, a small fraction of BAs can be uncoupled, dehydrogenated, and dehydroxylated by GM to form secondary bile acids [34,35,130]. This suggests that GM may play an important potential role in bile acid cycling.

Huang et al. [112] explored the regulatory effect of sulfated asparagus polysaccharide (GLP) on hyperlipidemia mice induced by a high-fat and high-cholesterol diet. The experiment found that GLP could reduce the hepatic total cholesterol (TC) and total triglyceride (TG) levels in the model group mice, and alleviate the accumulation of hepatic fat. Mechanistic analysis showed that GLP could accelerate the conversion of cholesterol from primary BAs to secondary BAs in the liver by promoting the expression of LxRα (liver X receptor α) and CYP7A1 genes involved in cholesterol metabolism and regulating the relative abundance of beneficial intestinal bacteria Bacteroides, Ruminococcus_1 and Lactobacillus involved in BAs metabolism. Not only that, GLP could increase the relative abundance of Prevotellaceae_UCG-001, Corprococcus_1, Alistipes and Roseburia, and the Lachnospiraceae_NK4A136_group, respectively, to reduce hydrophobic bile acids (e.g., chenodeoxycholic acid CDCA and DCA), and increase hydrophilic bile acid (e.g., GUDCA and TUDCA) levels, thereby reducing the accumulation of toxic bile acids (DCA, LCA, CDCA). Polysaccharides may improve liver injury by changing the ratio of different bile acids, thereby regulating the homeostasis of bile acids. There are also studies suggesting that changes in the ratio of bile acids may also increase FXR agonists (e.g., CDCA) and decrease its antagonists (e.g., CA, T-αMCA), thereby activating FXR-related pathways [131]. Li et al. [93] used *Grifola frondosa* heteropolysaccharide (GFP) as the treatment substance and NAFLD rats formed by HFD as the modelling group to study the improvement effect of GFP on NAFLD and explored the potential mechanism. Compared with the model group, the polysaccharide group could change the GM composition ratio, reduce F/B, significantly increase beneficial bacteria such as Allobaculum, Bacteroides and Bifidobacterium, and reduce the abundance of harmful bacteria such as Acetatifactor, Alistipes, Flavonifractor, Paraprevotella and Oscillibacter. In addition, the study also found that the gene expression of CYP7A1 was significantly increased after GFP intervention compared with the model group, and the expression of ACC, TNF-α and SOCS2 was effectively reduced. These results indicated that GFP could promote the conversion of hepatic cholesterol to BAs, improve cholesterol utilization, and reduce its accumulation to improve NAFLD. At the same time, it is suggested that GFP may play a certain role in promoting fat metabolism, reducing liver lipid levels and being anti-inflammatory. 

Regulation of bile acid cycle-related receptors (e.g., FXR) and their signaling pathway can affect the occurrence and development of NAFLD [44,47,48]. Polysaccharides can regulate the expression of FXR-related target genes, such as CYP7A1 (a key enzyme in the classical pathway of BAs synthesis), lipid metabolism-related SREBP1c and small intestinal brush border sodium-dependent bile acid transporter (ASBT, responsible for bile acid absorption), promote bile acid synthesis and excretion, and regulate lipids metabolism, thereby exerting an anti-NAFLD effect [93,132].

Zhong et al. [132] investigated the effect and mechanism of *Ganoderma lucidum* polysaccharide peptide (GLPP) (polysaccharide-peptide ratio of 95%: 5%) in improving NAFLD. GLPP could reverse the low expression of CYP7A1, FXR, SHP (small heterodimer partner) and other proteins and the high expression of FGFR4 in the liver of ob/ob mice and ApoC3 transgenic mice to regulate bile acid synthesis, and inhibit fatty acid synthesis by down-regulating the expression of SREBP1c, FAS, ACC, thereby ultimately improving NAFLD. During the bile acid cycle, most of the primary bile acids converted from hepatic cholesterol flow into the intestine through BSEP, then can be reabsorbed by intestinal ASBT, and enter the portal vein through OSTα/β. Finally, bile acids are taken up by NTCPs and OATPs into the liver [130]. Therefore, altering ASBT expression and activity can affect the extent of bile acid reabsorption [133,134]. Wang et al. [131] studied the improvement effect of inulin polysaccharide (inulin) on HFD-induced NAFLD mice and found that inulin could activate the FXR-FGF15 pathway and reduce the expression of ASBT, thereby increasing the de novo synthesis of hepatic bile acids (to reduce hepatic lipid deposition), inhibit intestinal absorption of bile acids and promote their efflux.

These studies suggested that polysaccharides could reduce the levels of harmful metabolites of gut flora (e.g., LPS), increase the type and abundance of beneficial metabolites (e.g., SCFAs), and regulate the related signaling pathways of the bile acid cycle through the flora, thereby playing an active role in anti-NAFLD. Figure 3 summarizes the potential mechanisms of polysaccharides in the prevention and treatment of NAFLD through the gut–liver axis and its related signaling pathways.

Polysaccharides can not only regulate the composition and proportion of intestinal flora, increase the diversity of flora and increase the abundance of beneficial bacteria, but also regulate flora metabolites and affect their related signaling pathways, thereby improving or preventing NAFLD.

(1) Polysaccharides regulate the composition and proportion of intestinal flora, such as reducing Firmicutes (F)/Bacteroidetes (B) ratio, promoting the colonization of beneficial flora (e.g., Lactobacillus, Bifidobacterium), and improving the microecology of intestinal flora, thereby play a role in prevention and treatment NAFLD.

(2) Polysaccharide inhibits the increase in LPS level, on the one hand, it reduces its stimulation to intestinal epithelial cells, reduces the inflammatory response of intestinal epithelial cells, reduces the damage of tight junction proteins, and improves intestinal barrier function; On the other hand, it reduces the entry of LPS into the liver through the portal vein and the production of LPS–LBP conjugates, which subsequently reduces the response of CD-14 and inhibits the activation of the TLR4 pathway, thereby reducing liver inflammation and relieving NAFLD (Figure 3–Pathway I).

(3) Polysaccharides act as prebiotics to increase SCFAs in the body. Such metabolites are not only beneficial to repairing the intestinal barrier and reducing harmful substances entering the liver, but also bind to receptors such as GPR41 and GPR43 in the liver after entering the blood through the portal vein, regulate the AMPK signaling pathway, increase lipid oxidation, decrease the expression of PPARγ and SREBP1c and reduce the synthesis and accumulation of hepatic lipid (Figure 3–Pathway II).

(4) Primary bile acids can be biotransformed to secondary bile acids under the action of GM. Polysaccharides affect bile acid metabolism and regulate the ratio between each bile acid. For example, polysaccharides can reduce FXR antagonist BAs (e.g., CA, T-αMCA) and increase FXR agonist BAs (e.g., CDCA); polysaccharides may also reduce the synthesis of cytotoxic BAs (e.g., DCA, LCA) and increase the synthesis of cytoprotective BAs (e.g., UDCA, TUDCA). The absorption of BAs into the liver can enhance the FXR-related signaling pathway, which not only induces the expression of the downstream target gene SHP, negatively regulates SREBP1c, and downregulates the expression of genes related to fatty acid syntheses, such as FAS and ACC, to reduce hepatic lipogenesis, but also reduces insulin resistance (IR) to ameliorate NAFLD (Figure 3–Pathway III).

(5) Bile acids can be synthesized by CYP7A1, transported by BSEP, and reabsorbed into the liver by ABST and OSTα/β. Polysaccharides may have a certain regulatory effect on the bile acid cycle and excretion. Polysaccharides can indirectly upregulate the activity of CYP7A1 through bile acid cycling, increase the conversion of liver cholesterol to bile acids, and increase the utilization of cholesterol; at the same time, up-regulate the expression of BSEP and down-regulate the expression of intestinal ASBT, which may promote bile acid transport and inhibit the reabsorption of bile acids by the intestinal wall, thereby reducing intrahepatic bile acid accumulation and resulting liver damage (Figure 3–Pathway IV). 

To conclude, the improvement effect of polysaccharides on NAFLD based on the gut–liver axis mainly reflects the characteristics of multiple pathways and multiple targets of the mechanism, but there are still many limitations in the existing research. For example, for the improvement of NAFLD, different polysaccharides may cause different biochemical and signaling changes. Therefore, the correlation between the improvement effect of each polysaccharide and its structural characteristics needs to be further explored. Second, many studies have shown that polysaccharides have an indirect improvement effect on NAFLD, but the underlying mechanism and signaling pathways are not fully understood. The reason may be that the pathogenesis of NAFLD based on the gut–liver axis has not been fully resolved, thus posing challenges to the study of polysaccharides. In this regard, more intensive studies need to be conducted to systematically understand the NAFLD-improving mechanism of polysaccharides based on the gut–liver axis. To better explore the potential medicinal value of polysaccharides, it is also necessary to study and confirm the effectiveness and druggability of polysaccharides in the prevention and treatment of NAFLD. Furthermore, carefully designed clinical trials (such as a double-blind, randomized, controlled multicenter trial) need to be carried out on the lead compounds of polysaccharides, which have not been reported yet.

## 4. Summary and Outlook

The incidence of NAFLD is increasing year by year, and long-term development can lead to liver cirrhosis and hepatocellular carcinoma. Moreover, NAFLD is closely related to chronic kidney disease, cardiovascular disease, and diabetes, all of which seriously threaten and affect public health and quality of life [2,135]. With the proposal of the gut–liver axis raised and its application in the study of NAFLD, the roles of GM microecology and gut barrier function in the pathogenesis and progression of NAFLD have been gradually uncovered. These studies undoubtedly provide a rationale for finding effective NAFLD intervention agents [19,48,136].

This article reviews the pathogenesis of NAFLD from the perspective of the gut–liver axis, and mainly expounds on its correlation with NAFLD from the perspective of intestinal flora imbalance and intestinal barrier permeability. Compared with healthy individuals, NAFLD patients have an imbalanced gut microbiome and impaired gut permeability. On the one hand, the beneficial metabolites of GM, such as SCFAs, are relatively reduced, while the proportion of harmful metabolites, such as LPS, is increased. LPS enters the portal vein through the damaged intestinal barrier, causing inflammation and energy metabolism disorders in the liver. At the same time, under the action of GM, the ratio of different types of bile acids in the body may be changed, which will eventually adversely affect the metabolism of glucose and lipids in the liver through enterohepatic circulation. On the other hand, impaired intestinal mucosal barrier function, increased intestinal permeability, immune imbalance, decreased IgA secretion, and inflammatory responses involving immune cells promote bacterial translocation and LPS entry into the blood, which further induces hepatic inflammatory responses. On this basis, the mechanism of action of polysaccharides in improving or preventing NAFLD by regulating the gut–liver axis was analyzed. On the one hand, polysaccharides can regulate the type and proportion of GM through the gut–liver axis, guide the body to change in a beneficial direction, improve NAFLD via maintaining the ecological balance of the flora and exerting a certain intestinal barrier repair function. On the other hand, polysaccharides can promote the production of SCFAs and the proliferation of beneficial bacteria, and inhibit the production of LPS and the abundance of harmful bacteria, whilst regulating bile acid-related signaling pathways to combat NAFLD, suggesting that polysaccharides can be used as a potential prebiotic to improve or prevent NAFLD. It is worth noting that the mechanism of NAFLD improvement based on the bile acid cycling pathway has not been fully elucidated, and the effect of polysaccharide intervention on bile acid homeostasis and its GM regulation is still poorly understood, so it needs to be further explored.

Studies have shown that traditional Chinese medicine has a significant effect on the treatment of NAFLD with fewer adverse reactions [137]. Polysaccharides, as a class of natural macromolecules widely present in traditional Chinese medicines (TCM), have attracted much attention due to their rich activities and reliable biosafety [138,139,140,141]. In recent years, it has been found that polysaccharides, especially from TCM, have the potential to prevent and treat NAFLD with diverse structures and mechanisms of action. Many TCM polysaccharides can inhibit the occurrence and development of NAFLD by maintaining GM ecological balance, regulating GM metabolites, and improving intestinal barrier function, suggesting that it may be feasible and promising to search for anti-NAFLD polysaccharides from TCM [93,97,100].

Although polysaccharides have potential anti-NAFLD effects, there are few studies on their clinical translation and practical application. Furthermore, their efficacy and druggability are still unclear. Moreover, compared with small molecules, polysaccharides are a class of macromolecular compounds with more complex structures, the related profound research has not yet been conducted. For example, the multi-target, multi-level mechanisms of action of polysaccharides and their relationships with structure have not been elucidated. Therefore, an in-depth study and understanding of the structure–activity relationship of active polysaccharides and more precise regulation of GM and repair of the intestinal barrier using the principle of the gut–liver axis will help to improve the therapeutic effects of polysaccharides on NAFLD and contribute to the discovery of a new generation of polysaccharide prebiotics. In addition, different types of GM may have different physiological and metabolic functions, and there will also be synergistic or competitive effects between bacterial groups. Revealing the intervention effect of polysaccharides based on the metabolic mechanism of GM will help to understand the mechanism of “specific polysaccharide–corresponding bacteria (species)–liver” interactions. It may also be more helpful in the development of more precise and targeted polysaccharide drugs.

## Figures and Tables

**Figure 1 ijms-23-11710-f001:**
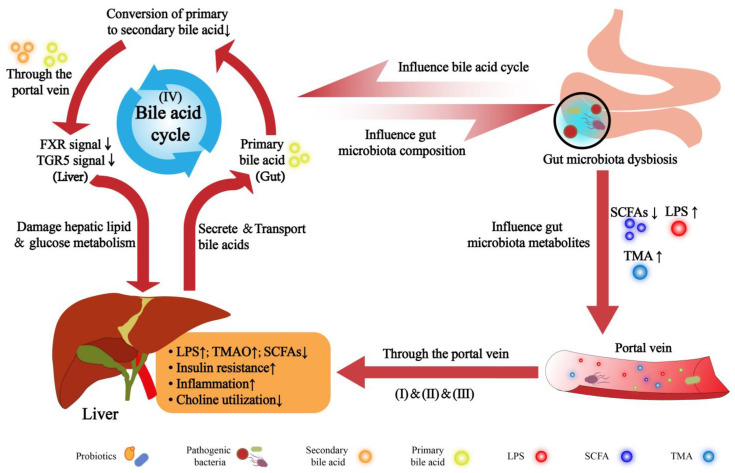
Intestinal dysbiosis is involved in the pathogenesis of the non-alcoholic fatty liver disease.

**Figure 2 ijms-23-11710-f002:**
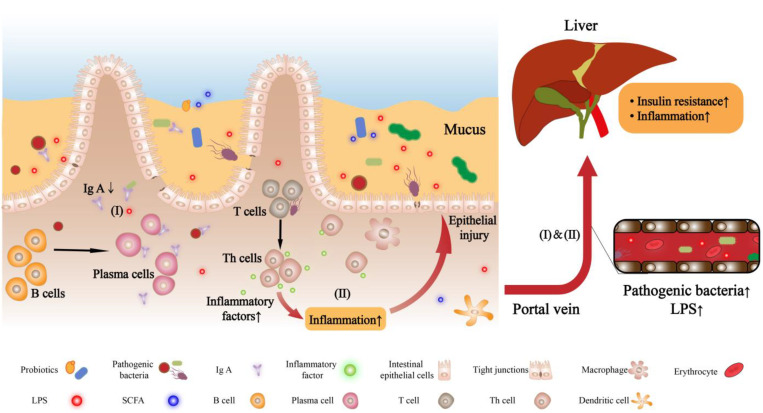
The pathogenesis of nonalcoholic fatty liver induced by changes in intestinal barrier permeability.

**Figure 3 ijms-23-11710-f003:**
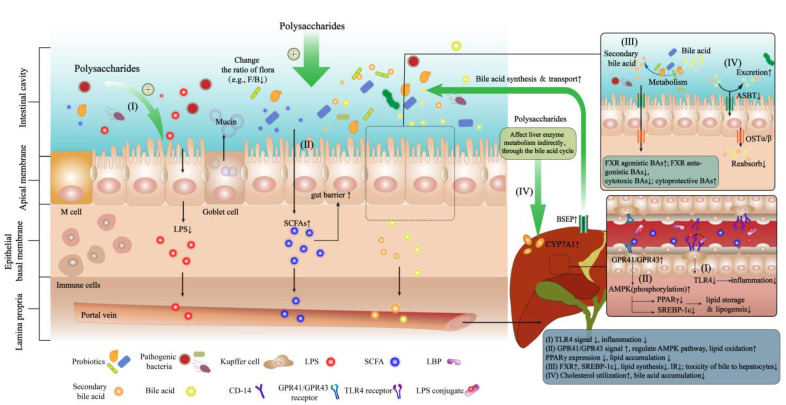
The potential mechanisms of polysaccharides in the prevention and treatment of NAFLD through the gut–liver axis.

**Table 1 ijms-23-11710-t001:** Effects of polysaccharides on alleviating NAFLD through the gut–liver axis.

Polysaccharides	Structural Information *	Dose	Disease Modelling Method	Related Biochemical Information, Signal Factor and Histological Change (Compared with the Model Group) ^#^	Variation of Gut Microbiota (Compared with the Model Group)	Reference
*Grifola frondosa* heteropolysaccharide (GFP)	(1) Mw:66.1 kDa (2) Monosaccharide composition:Fuc:Xyl:Man:Glc:Gal = 1.2:1.4:1.4:1.1:1.0 (3) Glycosidic linkage type:β-D-Glc*p*A→, 1,2,6-α-Gal, →2)-α-Man*p*→, →3)-α-L-Fuc*p*-(1→ [92]	150 mg/kg·bw	HFD for 8 weeks	(1) BW gain rates↓ (2) Attenuate liver damage on a histological level (3) AST, TG, TC, LDL-c↓; HDL-c↑ (4) ACC, TNF-α, SOCS2, CYP4A1↓; GSH-Px, SOD, CYP7A1↑	Firmicutes↓ Bacteroidetes↑ *Allobaculum, Bacteroides, Bifidobacterium, Blautia, Coprococcus, Phascolarctobacterium, Prevotella, Roseburia* ↑ *Acetatifactor, Alistipes, Flavonifractor, Paraprevotella, Oscillibacter* ↓	[93]
Noni fruit polysaccharide (NFP)	(1) Mw:456 kDa (2) Monosaccharide composition:GalA:Gal:Glc:Rha:Ara = 27.1:2.1:9.8:2.2:1.0 (3) Glycosidic linkage type:*O*-acetylated-(1→4)-α-GalA*p* (backbone), (1→2)-Rha*p*, (1→4)-β-Gal*p*, (1→5)/(1→ 3)-Ara [94]	100 mg/kg·bw	HFD for 4 weeks	(1) BW, LW, abdominal fat weight, spleen weight, BW gain rates↓ (2) Attenuate liver damage on a histological level (3) AST, ALT, TG, TC, LDL-c↓; HDL-c↑ (4) CCL5, TNF-α, IL-1β, liver IL-10, GPR43↓; MDA, hepatic TEAC, SOD, GSH-Px, CAT, ZO-1, occludin, total SCFA↑	Firmicutes↑ *Lactobacillus, [Eubacterium] coprostanoligenes group, Ruminococcus_1, Alloprevotella, Ruminococcaceae_UCG_014, Parasutterella, Muribaculaceae*↑ *Actinobacteria, Prevotella_9, Turicibacter*↓	[95]
Mussel polysaccharide α-*D*-glucan (MP-A)	(1) Mw:1.2 × 10^3^ kDa (2) Monosaccharide composition:Glc (3) Glycosidic linkage type:(1→4)-α-D-Glc, (1, 2)-α-D-Glc	600 mg/kg·bw	HFD for 3 weeks	(1) BW, BW gain rates↓ (2) Attenuate liver damage on a histological level (3) AST, ALT, TG, TC, LDL-L, ALP, plasma ethanol↓ (4) LPS, TNF-α, IL-1, IL-6, TLR4, NF-κB, SREBP-1c, PPARγ↓ (5) SCFAs↑	Firmicutes↓ Bacteroides↑ *Mailhella, Alloprevotella, Butyricimonas, Parabacteroides, Akkermansia, Bifidobacterium*↑ *Allobaculum, Pseudomonas, Hydrogenophaga, Romboutsia, Turicibacter, Ruthenibacterium, Faecalibaculum*↓	[96]
*Ophiopogon* polysaccharide (MDG-1)	(1) Mw:3.4 kDa (2) Monosaccharide composition:Fru, trace of GLc (3) Glycosidic linkage type:Fru*f* (2→1), Fru*f* (2→6) Fru*f* (2→)	HFD with MDG-1 (2‰, 4‰, 8‰)	HFD for 16 weeks	(1) BW↓ (2) Attenuate liver damage on a histological level (3) TG, TC, AST, ALT↓ (4) IL-1β, IL-4, TNF-α, CD68, SREBP-1c, FAS, ACC-1, PPARc↓; IL-10, AMPK, GPR41, GPR43↑ (5) SCFAs↑	Firmicutes/Bacteroidetes (F/B) ratio↓ *Lactococcus, Enterorhabdus, Turicibacter, Clostridium-sensu-stricto-1, Tyzzerella, Oscillibacter*↓ *Alistipes, Ruminiclostridium, Ricenella*↑	[97]
Inulin (INU)	(1) Mw:different degrees of polymerization (2-60 Fru units) (2) Monosaccharide composition:Glc, Fru (3) Glycosidic linkage type:(2→1)-β-D-Fru, α-Glu-(1→2)- β-D-Fru [98,99]	5000 mg/kg·bw	HFD for 14 weeks	(1) BW, LW, liver index↓ (2) Attenuate liver damage on a histological level (3) ALT, AST, TG, TC, plasma insulin, HOMA-IR↓ (4) F4/80^+^ Mψs, TLR4, NLRP3, ASC, caspase-1, NF-κB, IL-1β, IL-18, IL-6, LPS↓; IL-10↑ (5) SCFAs↑	F/B ratio↓ *Bififidobacterium, Akkermansia*↑	[100]
Sacran polysaccharide	(1) Mw:1–2.2 × 10^4^ kDa (2) Monosaccharide composition:Glc:Gal:Man:Xyl:Rha:Fuc:GalA:GlcA = 25.9:11.0:10.0:16.2:10.2:6.9:4.0:4.2 (traces of Ara, GalN, Muramic acid) (3) Glycosidic linkage type:→4)-6-deoxy acid/pentose-(1,4)-uronic acid-(1,4)-uronic acid--(1,4)-hexopyranose-(1,4)- hexopyranose-(1,4)-sulfated muramic acid-(1→ [101]	80 mg/kg·bw	HFC for 4 and 8 weeks	(1) BW, LW, liver index↓ (2) Attenuate liver damage on a histological level (3) TC, TG, AST, ALT, ALP↓ (4) TGF-β1, TNF-α↓	F/B ratio ↓ *Blautia*↑ *Prevotella*↓	[102]
*Astragalus* polysaccharides (APS)	Monosaccharide composition:Ara:Xyl:Glc:Gal:Rha = 1.0:14.6:0.5:44.1:2.2	HFD with APS (4% in finial concentration)	HFD for 14 weeks	(1) BW, LW, liver index↓ (2) Attenuate liver damage on a histological level (3) ALT, AST, TG, TC, serum insulin↓ (4) IL-1β, FASN, CPT-1α↓ (5) SCFAs, acetic acid level↑	The bacterial richness and evenness↑ F/B ratio↓ *Desulfovibrio vulgaris* (Traditional sulfate-reducing bacteria, producing acetic acid) ↑	[103]
*Sarcodon aspratus*, polysaccharides (SATP)	Monosaccharide composition:Gal:Ara:Man:Glc = 24.2:8.4:3.3:1.0	100, 200, 400 mg/kg·bw	HFD for 14 weeks	(1) Attenuate liver/ileum damage on a histological level (2) TC, TG, NEFA, ALT, AST↓; HDL-c↑ (3) LPO, MDA↓; SOD, GSH-Px↑ (4) IL-1β, TNF-α, LPS↓ (5) SCFAs ↑	F/B ratio↓ *Lactobacillus, Bacteroides, Akkermansia*↑	[104]
An insoluble polysaccharide from the sclerotium of *Poria cocos* (WIP)	(1) Mw:4.486 × 10^3^ kDa (2) Monosaccharide composition:Glc (3) Glycosidic linkage type:(1, 3)-β-D-Glc (main chain)	500, 1000 mg/kg·bw	ob/ob mice (8 weeks old)	(1) Attenuate liver damage on a histological level (2) AST, ALT, TC, TG, LDL-c↓; glucose tolerance insulin resistance↑ (3) LPS↓; SOD, Muc-5, ZO-1, Occludin, SCFAs ↑	*Lachnospiracea, Clostridium* IV*, Alloprevotella, Parabacteroides, Ruminococcus, Bacteroides*↑	[105]
Partially hydrolyzed guar gum (PHGG)	(1) Mw:1-100 kDa (2) Monosaccharide composition:Man:Gal = 2:1 (repeating unit of PHGG) (3) Glycosidic linkage type:(1,4)-β-D-Man, (1,6)-α-D-Gal [106]	Given an atherogenic diet with 5% PHGG	Atherogenic diet for 8 weeks; Administration of 0.5% DSS	(1) Attenuate liver damage on a histological level (2) AST, ALT, MPO activity↓ (3) TNF-α, Collagen 1a1, MCP-1, TLR4, TLR9, endotoxin levels in the portal vein, FD4 flux↓ (4) formic acid↑	*Bacteroides, Clostridium subcluster* XIVa↑ *Prevotella, Bifidobacterium, Clostridium cluster* XVIII↓ (Compare to control group)	[107]
Pectin	(1) Sub-domains composition:Homogalacturonan (HG, 65%), Rhammogalacturonan I (RG I, 20-35%), Rhamnogalacturonan II (<10%), Xilogalacturonan (<10%) (2) Glycosidic linkage type:partially methyl-esterified (1, 4)-α-D-GalA (HG), →4)-α-D-GalA-(1, 2)-α-L-Rha-(1→(RG I) [108,109]	2000 mg/kg·bw	HFD for 16 weeks	(1) TG, liver/body weight ratio↓ (2) Attenuate liver damage on a histological level	Firmicutes↓ Bacteroidetes↑ *Prevotellaceae*, *Turicibacteraceae*↑*Desulfovibrionaceae, Ruminococcus*↓	[110]
*Astragalus mongholicus* polysaccharides (mAPS)	Monosaccharide composition:Glc:Ara:Gal:Rib = 26.0:1.4:1.2:1.0	200 mg/kg·bw	HFD for 6 weeks	(1) BW, liver index, eWAT weight↓ (2) Attenuate liver damage on a histological level (3) AST, ALT, TG, TC, LDL-c, HOMA-IR↓ (4) ZO-1, occludin↑ (5) LPS, TLR4, NLRP3, TNF-α, IL-1β, NF-κB↓ (6) AMPK, SREBP-1, PPAR-α, GPR41, GPR43↓	F/B ratio↓ *Proteobacteria, Episilonbacteria* ↑	[111]
Sulfated *Gracilaria lemaneiformis* polysaccharide (GLP)	(1) Molecular weight:31.5 kDa (2) Monosaccharide composition:Gal:Glc:Fuc:Man = 9.2:6.6:1.0:0.6 (3) Glycosidic linkage type:→3,4)-Fuc*p*-(1→, →3,4,6)-Gal*p*-(1→, →4)-Glc*p*-(1→, →4,6)-Man*p*-(1→, →6)-Glc*p*-(1→, →6)-Gal*p*-(1→, Gal*p*-(1→	60, 225 mg/kg·bw	High-fat and high-cholesterol diet for 40 days	(1) BW, LW, epididymal fat weight↓ (2) Attenuate liver damage on a histological level (3) TG, TC, FFA↓ (4) TLCA, GDCA, GUDCA, TUDCA↑, AMPKα, CYP7A1↑; CA, CDCA, TCDCA, DCA↓, SBREP-2↓	F/B ratio↑ *Lachnospiraceae_NK4A136_group, Bacteroides, Ruminococcus_1, Lactobacillus*↑ *Prevotellaceae_UCG-001, Corprococcus_1, Alistipes, Roseburia*↑	[112]

Notes ***:** The structural information marked in the column is not from the same article as other content; *p* or *f* added after the monosaccharide indicates that the sugar is pyranose (*p*) or furanose (*f*); **#:** In the table, ‘↓’ means down-regulation after administration of polysaccharide; ‘↑’ means up-regulation after administration of polysaccharide.

## Data Availability

Not applicable.

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
