# Peer review of "Research Progress on the Therapeutic Effect of Polysaccharides on Non-Alcoholic Fatty Liver Disease through the Regulation of the Gut–Liver Axis"

_ijms, 2022, doi:10.3390/ijms231911710_

Round 1

Reviewer 1 Report

The authors aim to describe the potential positive effects of polysaccharides on pathogenesis and therapy of NAFLD.

In the first about 8 pages they describe the numerous possible mechanisms contributing to emergence of fat accumulation, such as intestinal dysbiosis, LPS, bile acids, SCFA, disturbed intestinal barrier function, among others. This is completely correct. However in many paragraphs there are only very few or no references. Therefore, is it is difficult to see, whether the described mechanisms correspond to accepted current knowledge or whether they are genuine ideas or speculations of the authors.

One of my main points of criticism is the use of the term “polysaccharides”. This encompasses a variety of compounds, compounds found in humans, plants, microorganisms, and the term polysaccharides is not further specified in the greatest part of the manuscript. Only later, we learn that the authors focus on several oligo- or polysaccharides originating from plants, fruits, bacteria. Only these compounds are described to exerts potential effects on the liver, as found in animal models.

It is e.g. true, that bacteria (bacterial [poly]-saccharides) can improve intestinal barrier function, that SCFA, mainly if not exclusively produced by intestinal bacteria, have positive effect on intestinal epithelial integrity, that GM bplays a role in bile acid metabolism.

However, what is the thread from “polysaccharides” as general term to NAFLD? Maybe from exogenous and not digestible (by humans) glucans to NAFLD?.

In view of these questions I suggest a complete different title of a manuscript. E.g.: “Modification of gut microbiota by exogenous glucans and their effect on NAFLD”. This furthermore means that I recommend not to publish the manuscript in the present form. However, I recommend that authors should restructure the complete manuscript and start a resubmission within due time.

Author Response

Reviewer #1:

Response to comment 1:

Point 1: In the first about 8 pages they describe the numerous possible mechanisms contributing to emergence of fat accumulation, such as intestinal dysbiosis, LPS, bile acids, SCFA, disturbed intestinal barrier function, among others. This is completely correct. However in many paragraphs there are only very few or no references. Therefore, is it is difficult to see, whether the described mechanisms correspond to accepted current knowledge or whether they are genuine ideas or speculations of the authors.

Response 1: Many thanks for the Reviewer’s question about the lack of references in the first 8 pages of our manuscript. Considering this comment, we carefully checked our original text again and found that there are about 50 references on “the possible mechanisms contributing to the emergence of fat accumulation” (pages 3-8, lines 115-350, in our original manuscript). In fact, up to the eighth page, we have cited a total of 88 references. Yet it’s worth noting that the legends for Figures in our original manuscript are indistinguishable from the text. This may cause these paragraphs in the manuscript to appear to have very few or no references. Other Reviewers have also raised a similar question about it. Accordingly, we reduced the character font of the legends to make it different from the main text. Finally, in the first 8 pages of our revised manuscript, we updated the citation with a total of 91 references, of which about 54 were about ‘numerous possible mechanisms contributing to the emergence of fat accumulation’ (pages 2-7, lines 72-309, in the revised manuscript). These references were published in recent years, which we believe can reflect the latest research progress on “The pathogenesis of NAFLD based on the gut-liver axis”.

Response to comment 2:

Point 2: One of my main points of criticism is the use of the term “polysaccharides”. This encompasses a variety of compounds, compounds found in humans, plants, microorganisms, and the term polysaccharides is not further specified in the greatest part of the manuscript. Only later, we learn that the authors focus on several oligo- or polysaccharides originating from plants, fruits, bacteria. Only these compounds are described to exerts potential effects on the liver, as found in animal models.

Response 2: We are very sorry for our negligence in the definition of “polysaccharides” in the previous paragraphs of our manuscript, which may lead to undeserved misunderstandings. In our manuscript, we proposed that polysaccharides are natural, non-starch substances, which mainly originated from plants, fungi and animals. They are not like LPS from special sources. So far, there are not many reports on the polysaccharides which can treat NAFLD via the gut-liver axis, thus a broad definition may be helpful for our citation and writing. In addition, in the original page 16 line 461, we have given the corresponding definition “The polysaccharides discussed in this article are different from ordinary starch, and they can play an important role in human health as prebiotic”.  Considering your and other Reviewers’ comments, we have made the following modifications.

 (1) The original content “The polysaccharides discussed in this article are different from ordinary starch, and they can play an important role in human health as prebiotics” (page 16, line 461) was modified as “The polysaccharides discussed here are different from ordinary starch or maybe called non-starch polysaccharides to some extent, which can play an important role in human health as prebiotics” and moved to page 2, lines 57-59 in our revised manuscript.

(2) The original content “Polysaccharides are biological macromolecules composed of 10 or more monosaccharides linked by glycosidic bonds……, providing researchers with different perspectives” (page 2, lines 64-73) was moved ahead to page 2, lines 46-54 of the revised manuscript.

(3) The original content “A variety of dietary polysaccharides in the human diet cannot be directly absorbed by the intestinal tract, and can only be utilized after being metabolized by a variety of GM enzymes” (page 1, lines 41-43) was modified as “A variety of dietary polysaccharides (e.g., plant polysaccharides, animal polysaccharides, fungal polysaccharides) in the human diet cannot be directly absorbed by the intestinal tract, and can only be utilized after being metabolized by a variety of GM enzymes” (page 2, lines 54-57) in our revised manuscript.

Response to comment 3:

Point 3: It is e.g. true, that bacteria (bacterial [poly]-saccharides) can improve intestinal barrier function, that SCFA, mainly if not exclusively produced by intestinal bacteria, have positive effect on intestinal epithelial integrity, that GM bplays a role in bile acid metabolism. However, what is the thread from “polysaccharides” as general term to NAFLD? Maybe from exogenous and not digestible (by humans) glucans to NAFLD?

Response 3: Thank you for your question about the thread from “polysaccharides” to NAFLD. First of all, the polysaccharides discussed in our manuscript are not polysaccharides of special origin, but natural, non-starch polysaccharides with a wide range of sources derived from plants, fungi, animals (such as Marine organisms, mussel polysaccharides MP-A) as mentioned in our manuscript. These polysaccharides are exogenous polysaccharides that are not easily digested by humans, including but not limited to the glucans you have mentioned. Second, our manuscript aims to state the potential role of polysaccharides in improving NAFLD through the gut-liver axis. To make the overall idea of the manuscript smooth and clear, we have made the following modifications.

(1) Clarify the definition of the polysaccharides discussed in the manuscript (page 2, lines 57-59, in the revised manuscript);

(2) Optimize the structure of the manuscript to make the idea of the manuscript clearer. We have restructured the manuscript by fusing the second and fourth sections in the original manuscript as a single section after the third section. A link of polysaccharides with NAFLD can be achieved in that the pathogenesis of NAFLD is based on the gut-liver axis, and the gut microbiota and its metabolites (such as LPS, SCFAs, BAs) communicate with the liver through the intestinal barrier and affect the occurrence and development of NAFLD. The polysaccharides can play a role in improving NAFLD in various ways (such as anti-inflammatory, anti-apoptosis, and anti-oxidative stress), while our manuscript focus on the natural, exogenous, non-starch polysaccharides and their indirect effects on NAFLD through the gut-liver axis. The latter may include maintaining the ecological balance of the flora and regulating the metabolites of the flora (such as LPS, SCFAs, BAs). In this regard, polysaccharides may be used as a potential prebiotic to improve or prevent NAFLD.

Response to comment 4:

Point 4: In view of these questions I suggest a complete different title of a manuscript. E.g.: “Modification of gut microbiota by exogenous glucans and their effect on NAFLD”. This furthermore means that I recommend not to publish the manuscript in the present form. However, I recommend that authors should restructure the complete manuscript and start a resubmission within due time.

Response 4: Thank you for your constructive comments. According to the above explanation, the definition of polysaccharides we introduced includes but is not limited to exogenous glucans. Therefore, we think that the term “exogenous glucans” may not be appropriate for our manuscript, and we preferred to remain the original title, but just making a minor modification by removing ‘the’ before the term ‘NAFLD’. As for the comment about “restructure the complete manuscript”, we moved the second section of the original manuscript (pages 2-3, lines 56-114, in the original manuscript, entitled ‘Polysaccharides and NAFLD’) to the 4th section (pages 8-9, lines 351-629, in the original manuscript, entitled ‘Ameliorative effects of polysaccharides on NAFLD via the gut-liver axis’), re-organizing it to be the first part of the new 4th section. The title of this new section is ‘3.1 Ameliorative effect of polysaccharides on NAFLD’ in the revised manuscript (pages 7-8, lines 310-332). In addition, we checked the whole text content carefully when moving and merging paragraphs, and deleted the repeated content and improved any incorrect expressions.

Reviewer 2 Report

COMMENTS

1.  I found a significant amount of relevant references related to the gut-liver axis and in it´s relationship to NAFLD that are missing (e.g. https://doi.org/10.1038/s12276-021-00614-x or https://doi.org/10.1038/s41575-020-0269-9 among many others). I am aware of the fact that this is not the main scope of the review but I appreciate important concepts and ideas in the field that are not being included nor appropriately discussed. I strongly encourage authors to consider reading them and citing them when appropriate. There are some references, in addition, which would benefit from an updating (e.g. ref 45).

2.     I encourage authors to consider fusing sections 2 and 4 and including them as a single section after section 3.

3.     Reviews should avoid including data based on PhD dissertations (≈10) or in journals not accessible to all readers, as readers will not be able to check the quality of this information.

4.Spelling and grammar mistakes are frequent throughout the text. Figure footnotes are included throughout the text, which makes the reading of the review quite confusing.

Author Response

Reviewer #2:

Response to comment 1:

Point 1: I found a significant amount of relevant references related to the gut-liver axis and in it´s relationship to NAFLD that are missing (e.g. https://doi.org/10.1038/s12276-021-00614-x or https://doi.org/10.1038/s41575-020-0269-9 among many others). I am aware of the fact that this is not the main scope of the review but I appreciate important concepts and ideas in the field that are not being included nor appropriately discussed. I strongly encourage authors to consider reading them and citing them when appropriate. There are some references, in addition, which would benefit from an updating (e.g. ref 45).

Response 1: Many thanks for your constructive comments. The references the Reviewer provided are very valuable and meaningful. After careful reading, we found that they can be cited in our manuscript and truly help express the relevant content we want to explain. Considering the Reviewer’s suggestion, we have further checked our manuscript carefully and found some other outdated references. Therefore, we made the following modifications.

(1) The first recommended reference entitled “The interaction between the gut microbiota and dietary carbohydrates in non-alcoholic fatty liver disease” mainly reviewed the role of dietary carbohydrates (glucose, fructose, and fiber) and gut microbiota in NAFLD. This reference mentioned “Sequencing studies indicated specific changes in microbiota composition in NAFLD subjects. Healthy versus obese/NAFLD individuals showed marked population differences in members of the phyla Bacteroidetes and Firmicutes”. The reference also showed that a reduction in microbiome diversity, rather than specific microbial populations may contribute to NAFLD, revealing the mechanism of intestinal flora imbalance leading to NAFLD. We cited it on page 3, line112 of our revised manuscript, and the serial number is 29.

(2) The second recommended reference entitled “Gut microbiota and human NAFLD: disentangling microbial signatures from metabolic disorders” described the role of intestinal flora metabolites (such as LPS, SCFA, BA, TMAO) in the occurrence and development of NAFLD. We cited it on page 3, line 95 of the revised manuscript, and the serial number is 24.

(3) Page 3 and line 146 in the original manuscript cited an outdated reference with serial number 45, so we replaced it with two more appropriate references. The original statement about “The GM mainly consists of five phyla: Firmicutes (79.4%), Bacteroidetes (16.9%), Actinobacteria (2.5%), Proteobacteria (1%) and Verrumicrobia (0.1%)” was updated as “More than 90% of the bacterial components in GM are mainly composed of four phyla: Firmicutes, Bacteroidetes, Actinobacteria, and Proteobacteria” (page 3, line 103, the reference numbers marked as 25 and 26, in the revised manuscript).

(4) Page 5 and line 211 in the original manuscript cited an outdated reference with serial number 58. We replaced it with a more appropriate one (page 4, line 169, the reference number marked as 42, in the revised manuscript).

(5) Page 5 and line 218 in the original manuscript cited an outdated reference with serial number 62. We replaced it with a more appropriate one (page 4, line 176, the reference number marked as 46, in the revised manuscript).

Response to comment 2:

Point 2: I encourage authors to consider fusing sections 2 and 4 and including them as a single section after section 3.

Response 2: We thank the Reviewer’s constructive comment. According to this comment, we have made the following changes.

 (1) Combined the second section of the original manuscript (pages 2-3, lines 56-114, entitled ‘Polysaccharides and NAFLD’) to the 4th section (pages 8-9, lines 351-629, in the original manuscript, entitled ‘Ameliorative effects of polysaccharides on NAFLD via the gut-liver axis’), making it the first part of the new 4th section. The title of this new section is ‘3.1 Ameliorative effect of polysaccharides on NAFLD’ in the revised manuscript (pages 7-8, lines 310-332). We hope the overall structure of the revised manuscript has been well-organized.

(2) Furthermore, we carefully checked the manuscript content when moving and merging paragraphs, deleted the repeated contents and revised the incorrect expressions.

Response to comment 3:

Point 3: Reviews should avoid including data based on PhD dissertations (≈10) or in journals not accessible to all readers, as readers will not be able to check the quality of this information.

Response 3: We are sorry for our negligence in the problem with the quality of the references. This comment does not doubt be important guiding significance. Considering the Reviewer’s suggestion, our revision idea is to replace the theses/dissertations with the published papers by the original authors. For those unpublished, we choose to replace it with a new reference, and the original content may have been partially modified. In addition, some references in journals not accessible to all readers (e.g., some Chinese journals) may be of a bit poor quality, which we also replaced and revised. As we have added two new references (numbered 24 and 29 in the revised manuscript) according to ‘comment 1’, the total number of references increased by 2 to 141. Thus, the new list of references has been generated as seen in the revised version.

Specific modifications:

(1) Page 2 and line 88 in the original manuscript cited a dissertation with serial number 33, so we replaced it with an article published by the authors (page 7, line 317, the reference number marked as 86, in the revised manuscript), and deleted the sentences about "Qu et al. studied the hepatoprotective effect of the component PNP40c-1 in pinus koraiensis pine nut polysaccharide ......, thereby alleviating the inflammatory response”.

(2) Page 3 and line 108 in the original manuscript cited a thesis with serial number 37, which had no published papers. We thus replaced it with other reference, and the original statements (e.g., “Inonotus obliquus polysaccharides could significantly reduce the body weight and liver coefficient of HFD-induced NAFLD mice……had biological activities of reducing blood lipids, preventing atherosclerosis, and protecting the liver”) were removed and modified as “Lentinan can improve NAFLD by activating the peroxisome proliferator-activated receptor alpha (PPAR-α) pathway to improve hepatic steatosis and reduce oxidative stress and apoptosis” (page 8, line 327, the reference number marked as 90, in the revised manuscript).

(3) Page 3 and line 112 in the original manuscript cited a thesis with serial number 38, which had no published papers. We thus replaced it with another reference, and the original statements (e.g., “It had also been reported that stevia root polysaccharides could improve NAFLD rats through the antioxidant effects, whose mechanism…… protein expression” were removed and modified as “It has also been reported that the hepatoprotective effect of Lycium barbarum polysaccharides on NAFLD is mainly reflected in the promotion of mitochondrial biogenesis, regulation of energy balance, reduction of intracellular lipid accumulation, and reduction of oxidative damage in NAFLD cell models” (page 8, line 330, the reference number marked as 91, in the revised manuscript).

(4) Page 15 and line 413 in the original manuscript cited a thesis with serial number 117. We replaced it with an article published by the authors (page 14, line 398, the reference number marked as 96, in the revised manuscript).

(5) Page 6 and line 257 in the original manuscript cited a Chinese reference with serial number 72. This reference may not be accessible to all readers, so we replaced it with another two appropriate references with better quality. The original statements about “The type and concentration of SCFAs in the intestinal and portal circulation …...and the proportion of butyrate decreased more significantly” were removed and modified as “The type and concentration of SCFAs in the intestinal and portal circulation may also be closely related to the progression of NAFLD. One of the studies showed ……. Another research found …… slowing the progression of NAFLD disease” (page 5, lines 213 and 221), the reference numbers were marked as 55 and 56, respectively, in the revised manuscript.

(6) Page 15 and line 451 in the original manuscript cited a Chinese reference with serial number 122. This reference may also not be accessible to all readers, so we replaced it with another appropriate reference as well. The original statements about “Most SCFAs pass through ……where they are utilized by various cells” were removed and modified as “A small amount of butyrate with other SCFAs pass through the intestinal epithelium into the portal vein and enter the circulation, where they are utilized by various cells” (page 14, line 436), the reference number was marked as 122 in the revised manuscript.

Other changes (for a similar reason to above ‘(5) and (6)’):

(1) The Chinese reference with serial number 32 in the original manuscript (page 2, line 80) was changed to a new reference with a similar statement in the revised manuscript (page 7, line 317, the reference number marked as 85).

(2) The Chinese reference with serial number 39 in the original manuscript (page 3, line 123) was changed to a new reference with a similar statement in the revised manuscript (page 2, line 80, the reference number marked as 19).

(3) The Chinese reference with serial number 43 in the original manuscript (page 3, line 138) was changed to a new reference with a similar statement in the revised manuscript (page 3, line 95, the reference number marked as 24).

(4) The Chinese reference with serial number 65 in the original manuscript (page 5, line 230) was changed to a new reference with a similar statement in the revised manuscript (page 5, line 188, the reference number marked as 48).

(5) The Chinese reference with serial number 86 in the original manuscript (page 7, line 327) was changed to a new reference with a similar statement in the revised manuscript (page 7, line 287, the reference number marked as 71).

(6) The Chinese reference with serial number 135 in the original manuscript (page 20, line 666) was changed to a new reference with a similar statement in the revised manuscript (page 19, line 649, the reference number marked as 137).

(7) The Chinese reference with serial number 6 in the original manuscript (page 1, line 43) was changed to a new reference with a similar statement in the revised manuscript (page 2, line 57, the reference number marked as 10).

(8) The Chinese reference with serial number 101 in the original manuscript (page 15, line 435) was changed to a new reference with a similar statement in the revised manuscript (page 14, line 420, the reference number marked as 117).

As for the problem of the references in Table 1, we have given the response to comment 3 of Reviewer 3. The main modifications are listed as follows.

(1) 8th and 11th rows: Two rows citing three theses/dissertations in the original Table 1 were deleted due to the data being probably inaccessible to all readers. The first one deleted was about “SATP” (Sarcodon aspratus polysaccharides) and cited in the 8th row with the reference number 101 in the original manuscript, and replaced by a published paper by the relevant authors (8th row, the reference number 104 in revised version). The second one was about “Yupingfeng polysaccharide” and was cited in the 11th row with the references (number 104 and 105) in the original version, which was deleted directly due to no other alternative reference.

(2) 10th, 13th and 16th rows: Although the 10th row (the reference number 103), 13th row (the reference numbers 107 and 108) and the 16th row (the reference number 112) in original Table 1 gave some information on a partial role of polysaccharides (that is GLPP, PCP and INU, respectively) in ameliorating NAFLD, however, the related references did not show any certain effects of polysaccharides on NAFLD through gut microbiota. These references may not reflect more comprehensively the mechanism of the gut-liver axis compared with others in Table 1. Therefore, we have put them out of Table 1 in our revised manuscript.

(3) Other changes in the original Table 1: Deleted the reference (3rd row, the reference number 93 in original version) because another reference in the same row (original number 17, revised number 96) has suggested the structural information of polysaccharides; Deleted the reference (7th row, the reference number 99 in original version) because another reference in the same row (original number 100, revised number 103) has suggested the structural information of polysaccharides; Deleted the reference (5th row, the reference number 95 in original version), and changed to two new references (5th row, the reference numbers 98 and 99 in the revised version), because their information could be accessible to all readers; Deleted the reference (14th row, the reference number 109 in original version), and changed to two new references (11th row, the reference numbers 108 and 109 in the revised version), because their information could be accessible to all readers; Added a new reference (10th row, the reference number 106 in the revised version) for structural information because the original reference (original number 106, revised number 107) on the polysaccharides with the same name and from the same source provided little structural information. 

Response to comment 4:

Point 4: Spelling and grammar mistakes are frequent throughout the article. Figure footnotes are included throughout the article, which makes the reading of the review quite confusing.

Response 4: We are very sorry for the many spelling and grammatical errors and the problem with the ‘Figure footnotes’ present in our manuscript. According to the Reviewer’s comments, we have worked on both language and readability and tried our best to make corrections to these questions. Firstly, we have carefully checked the spelling and grammar errors in the whole manuscript and revised them. Moreover, we have changed the font of the figure footnotes into a smaller one compared with the main text body, aiming to separate these two parts. These changes could be tracked in the revised manuscript. We hope that these modifications would make the manuscript easier to read and meet with approval. We will be happy to edit the text further, based on helpful comments from the reviewers.

Reviewer 3 Report

The authors present a narrative review of the published literature on noted progress of polysaccharides in improving nonalcoholic fatty liver disease (NAFLD) through the gut-liver axis.

This is a somewhat interesting review on what is an emerging worldwide major health problem. The review is well-structured and the figures are illustrative. A number of issues to be addressed by the authors:

- Be consistent with the abbreviations throughout the text ie. NAFL later becomes NAFLD, which is not primarily abbreviated

- Figures need to be larger as in the current form their resolution is poor

- Table 1 is too lengthy and hard to read, equally needs to be more uniformal ie. structural information is not uniformally presented in the different polyssacharides

- A limitations section needs to be more clearly provided, before the summary/conclusion section.

- Are there any ongoing RCTs? could be interesting to provide more information if available 

Author Response

Reviewer #3:

Response to comment 1:

Point 1:  Be consistent with the abbreviations throughout the article ie. NAFL later becomes NAFLD, which is not primarily abbreviated

Response 1: Many thanks for the Reviewer’s comment. We have checked the abbreviations in our manuscript again and found that the abbreviation of non-alcoholic fatty liver disease is NAFLD, which is consistent in the full manuscript. The NAFL mentioned in our manuscript is the abbreviation of non-alcoholic fatty liver, which is different from NAFLD. When NAFL was mentioned in the first place (page 1, line 30) of the manuscript, there are statements that “Non-alcoholic fatty liver disease, which mainly refers to a clinicopathological syndrome characterized by lipid accumulation and inflammation, is caused by multiple factors (e.g., insulin resistance, genetic predisposition, improper diet, obesity), except by alcohol and other certain liver damage factors (physical, viral, chemical liver damage). It includes non-alcoholic fatty liver (NAFL), non-alcoholic steatohepatitis (NASH) and related liver cirrhosis and hepatocellular carcinoma”, indicating that NAFL is a stage in the disease process of NAFLD. The original reference we cited (page 1, line 31, the reference number marked as 1) also mentioned that “NAFLD is subdivided into two primary subtypes, non-alcoholic fatty liver (NAFL) and non-alcoholic steatohepatitis (NASH), which is considered the progressive form of NAFLD”. Therefore, the abbreviation for NAFL appeared only in two places in our manuscript, which refer to the definition (page 1, line 30) and the disease process (page 3, line 95) of NAFLD, respectively, in our revised manuscript. The other parts of the manuscript just use the abbreviation of NAFLD.

Response to comment 2:

Point 2: Figures need to be larger as in the current form their resolution is poor

Response 2: We agree with this suggestion and have modified the Figures in resolution and display area in our manuscript. We used adobe AI software to output the image into TIF format, and adjust the resolution from 300 dpi to 600 dpi. Then, we inserted the uncompressed Figures into the manuscript with high fidelity. We hope this revision will meet with approval.

Response to comment 3:

Point 3: Table 1 is too lengthy and hard to read, equally needs to be more uniformal ie. structural information is not uniformally presented in the different polyssacharides.

Response 3: We truly agree with your comment on Table 1, which plays an important role in our manuscript. We have checked all the relevant references and tried to make the format of the polysaccharide structure information to be consistent. Besides, we have reduced long expressions in the Table. Moreover, we have removed the references with little information on the gut-liver axis (e.g., 10th, 13th and 16th row in original Table 1). It would be mentioned that in some references we cited in Table 1, there were mainly the pharmacological experiments of polysaccharides against NAFLD, but little or no polysaccharide structure information; even the structure of the polysaccharides from the same source was studied in some other references, but the results were usually inconsistent with each other. Therefore, it is difficult to achieve complete consistency in both content and format. Thus, we approach it in the following way. For the selected references on the polysaccharide pharmacology that have provided the structure information of polysaccharides, we would not cite other references, while for those without the structure information, we would search for related references on polysaccharides with the same name and from the same source and add the structural information as supplements.

Specific modifications:

(1) 3rd and 4th columns in original Table 1

The contents of the "Dose" and "Disease modelling method" columns have been simplified. We preserve important information such as polysaccharide concentration, main modelling method and dosing time while deleting the proportion of modelling diet, the administering method, etc.. For example, “Given HFD with daily gavage of GFP 150 mg/kg ·bw” was changed to “150 mg/kg ·bw”; “Fed with an HFD (67% control diet (13.5% energy from fat), 20% sucrose, 10% lard, and 3% cholesterol) for 8 weeks” was changed to “HFD for 8 weeks”.

(2) 2nd column in original Table 1

For the format of "Structural information", we mainly unify into three items: (1) Mw; (2) Monosaccharide composition; (3) Glycosidic linkage type.  In Table 1, the structure information format of GFP, NFP, MP-A, MDG-1, INU, Sacran, WIP, PHGG, and GLP have been modified and are generally consistent. In the references about APS, SATP and mAPS, the structure information was given but not complete. To avoid confusion with other literature, we did not cite more references again. In addition, the structure of pectin was complex, so it was present in another format.

(3) 8th and 11th rows in original Table 1

Two rows citing theses/dissertations in Table 1 were deleted due to the data probably being inaccessible to all readers. The first one deleted was about “SATP” (Sarcodon aspratus polysaccharides) and cited in the 8th row with the reference number 101 in the original manuscript, and replaced by a published paper by the relevant authors (8th row, the reference number 104 in revised version). The second one was about “Yupingfeng polysaccharide” and was cited in the 11th row with the reference numbers 104 and 105 in the original version, which was deleted directly due to no other alternative reference.

(4) 10th, 13th and 16th rows in original Table 1

Although the 10th row (the reference number 103), 13th row (the reference numbers 107 and 108) and the 16th row (the reference number 112) in original Table 1 gave some information on a partial role of polysaccharides (that is GLPP, PCP and INU, respectively) in ameliorating NAFLD, however, the related references did not show any certain effects of polysaccharides on NAFLD through gut microbiota. These references may not reflect more comprehensively the mechanism of the gut-liver axis compared with others in Table 1. Therefore, we have put them out of Table 1 in our revised manuscript. We hope that such a modification could ensure the uniformity of the Table structure.

Other changes in Table 1

(1) Deleted the reference (3rd row, the reference number 93 in the original version) because another reference in the same row (original number 17, revised number 96) has suggested the structural information of polysaccharides.

(2) Deleted the reference (7th row, the reference number 99 in the original version) because another reference in the same row (original number 100, revised number 103) has suggested the structural information of polysaccharides.

(3) Deleted the reference (5th row, the reference number 95 in the original version), and changed to two new references (5th row, the reference numbers 98 and 99 in the revised version), because their information could be accessible to all readers.

(4) Deleted the reference (14th row, the reference number 109 in the original version), and changed to two new references (11th row, the reference numbers 108 and 109 in the revised version), because their information could be accessible to all readers.

(5) Added a new reference (10th row, the reference number 106 in the revised version) for structural information because the original reference (original number 106, revised number 107) on the polysaccharides with the same name and from the same source provided little structural information.

Response to comment 4:

Point 4: A limitations section needs to be more clearly provided, before the summary/conclusion section.

Response 4: We agree with this good suggestion, which has important implications for the structure of our manuscript. We have added one paragraph before the ‘Summary and Outlook’ section, which can be seen on page 18, lines 595-611 in our revised manuscript.

Response to comment 5:

Point 5: Are there any ongoing RCTs? could be interesting to provide more information if available 

Response 5: Thank you for your question about RCTs, which is really important issue for the development of promising polysaccharide drugs against NAFLD. We have checked the current literature about the possible randomized controlled trials on the treatment of NAFLD with polysaccharides but get no hits, thus we include it as a portion of the limitations (please refer to ‘Response 4’ ) in our revised manuscript. We think it will be very interesting and important to research the clinical translation and practical application of the NAFLD-improving polysaccharides in the future.

Round 2

Reviewer 2 Report

The new version of the paper addresses most of my previous comments.

Reviewer 3 Report

The authors have critically revised their manuscript, now suitable for publication.